# MEC/Cloud Orchestrator to Facilitate Private/Local Beyond 5G with MEC and Proof-of-Concept Implementation

**DOI:** 10.3390/s22145145

**Published:** 2022-07-08

**Authors:** Jin Nakazato, Zongdian Li, Kazuki Maruta, Keiichi Kubota, Tao Yu, Gia Khanh Tran, Kei Sakaguchi, Soh Masuko

**Affiliations:** 1Tokyo Institute of Technology, Tokyo 152-8552, Japan; lizd@mobile.ee.titech.ac.jp (Z.L.); maruta.k.aa@m.titech.ac.jp (K.M.); kubota.k.aj@m.titech.ac.jp (K.K.); yutao@mobile.ee.titech.ac.jp (T.Y.); khanhtg@mobile.ee.titech.ac.jp (G.K.T.); 2Rakuten Mobile, Inc., Tokyo 151-0051, Japan

**Keywords:** Beyond 5G, virtualization, multi-access edge computing, cloud, private/local telecom operator, MEC B5G orchestrator, MEC proof-of-concept

## Abstract

The emergence of 5G-IoT opens up unprecedented connectivity possibilities for new service use cases and players. Multi-access edge computing (MEC) is a crucial technology and enabler for Beyond 5G, supporting next-generation communications with service guarantees (e.g., ultra-low latency, high security) from an end-to-end (E2E) perspective. On the other hand, one notable advance is the platform that supports virtualization from RAN to applications. Deploying Radio Access Networks (RAN) and MEC, including third-party applications on virtualization platforms, and renting other equipment from legacy telecom operators will make it easier for new telecom operators, called Private/Local Telecom Operators, to join the ecosystem. Our preliminary studies have discussed the ecosystem for private and local telecom operators regarding business potential and revenue and provided numerical results. What remains is how Private/Local Telecom Operators can manage and deploy their MEC applications. In this paper, we designed the architecture for fully virtualized MEC 5G cellular networks with local use cases (e.g., stadiums, campuses). We propose an MEC/Cloud Orchestrator implementation for intelligent deployment selection. In addition, we provide implementation schemes in several cases held by either existing cloud owners or private and local operators. In order to verify the proposal’s feasibility, we designed the system level in E2E and constructed a Beyond 5G testbed at the Ōokayama Campus of the Tokyo Institute of Technology. Through proof-of-concept in the outdoor field, the proposed system’s feasibility is verified by E2E performance evaluation. The verification results prove that the proposed approach can reduce latency and provide a more stable throughput than conventional cloud services.

## 1. Introduction

Beginning in 2019, fifth-generation mobile communication systems (5G) have been commercialized worldwide [1,2]. However, current leading services are primarily driven by 3G/4G-enabled smartphone platforms. The extraordinary features of 5G, such as ultra-high throughput, have not been fully leveraged [3]. No de facto service or scenario has been demonstrated in 5G mobile communications, including enhanced Mobile Broadband (eMBB), massive Machine Type Communications (mMTC), or Ultra-Reliable and Low Latency Communications (URLLC), as defined by International Telecommunication Union (ITU-R) M.2083 [4] in 2015. Therefore, mobile communication companies around the world are scrambling to ship 5G services in various areas and release more functions gradually.

Meanwhile, the shift from Mobile Virtualized Network Operator (MVNO) to Mobile Network Operator (MNO) is progressing [5], and new operators [6,7] are being established in markets dominated by the existing mobile network operators. In this trend, virtualization, which can support everything from Radio Access Networks (RAN) to applications, helps to quickly provide service at low cost. As a result, various operators [8] can more easily start and provide mobile services. Thanks to innovation in virtualization technologies, edge computing enables third-party applications to access network/computing/disk resources in resource pools without being aware of their physical locations. As its name implies, edge computing utilizes resources in proximity to users. The European Telecommunications Standards Institute (ETSI) has defined Multi-access Edge Computing (MEC) as able to support mobile networks as well as fixed/WiFi networks [9,10,11]. With the assistance of near-site computing resources, MEC can handle large amounts of mobile user data and alleviate traffic load on the backhaul [12,13,14], as shown in Figure 1. Beginning with the 4G era various consortia and organizations of interest have been devoting efforts to promote MEC, as seen in many demonstration experiments and press releases [15,16,17,18,19]. However, no practical service has been delivered yet. These discussions remain ongoing while 5G service has started. As stated in [20,21,22,23], new infrastructure is required in order for MEC to take off the MEC, as current mobile networks are not very compatible with virtualization technology. In addition, key use cases are eagerly awaited, and management and operation strategies for MEC applications should be clarified.

On the other hand, the 3rd Generation Partnership Project (3GPP) has involved MEC as local data networks in the architecture design from Release 15 [24]. It defined the N6 interface to associate MEC with the User Plane Function (UPF) of the 5G Core (5GC) and designed a local breakout for data traffic routing. Moreover, in globally published white papers on Beyond 5G (B5G), MEC employing virtualization technology has been acknowledged as one key enabler and an essential architectural network component.

In previous studies, we have focused on “who will use MEC (whether and how)”. Specifically, we first proposed a new Private/Local Operator to deploy MEC and analyzed the number of MEC that could maximize revenue [25,26]. In order to consider the impact of MEC on other operators, Private/Local Operator and Cloud Owner were discussed and analyzed using game theory [27,28]. The above results allowed us to establish an MEC ecosystem. However, what remains is the uncertainty of MEC management, which will affect operators’ operational decision-making about how to manage the life cycle of third-party applications in MEC. In this light, this paper builds on previous work [28], proposes a detailed MEC/Cloud Orchestrator to make it work, and provides a PoC implementation of the E2E system. Specifically, the contributions of this paper are unique and distinctive in two aspects. First, we clearly explain the definition of players related to MEC and each player’s role therein. Then, we discuss orchestrators’ issues, which have not been discussed among the players, and propose a new MEC/Cloud Orchestrator architecture. This paper additionally proposes a deployment method to enhance the feasibility of MEC applications within this architecture. This deployment method is discussed and detailed in two cases. The first case is that the Cloud Owner has the MEC/Cloud Orchestrator, and the second case is that the Cloud Owner and Private/Local Operator have the MEC/Cloud Orchestrator divided into function levels. The implementation method is designed for each case, and the advantages and disadvantages are discussed. From the end user’s perspective, these implementation methods, including SDN/NFV, improve Quality of Service (QoS) and Quality of Experience (QoE), as SDN/NFV contributes to flexible application deployment based on user locations as well as enhanced scalability of E2E network connections [18]. Next, we design the entire system to verify the superior performance of the proposed architecture. The implemented system is deployed as an edge cloud at Ookayama Campus of Tokyo Institute of Technology. A Proof-of-Concept (PoC) testbed for Beyond 5G was constructed by installing and deploying radio units outdoors. The testbed was built with State-of-the-Art (SOTA) production hardware (e.g., 5G, Sub-6/mmWave, fully virtualized). The proposed system with MEC is validated in this PoC field. The verification results suggest the achievement of lower-latency services and further proves that stable communication is enabled. This demonstration provides our readers with an example of the uniqueness of the proposed architecture.

The remainder of the paper is organized as follows. Section 2 introduces related works in order to help readers understand SOTA MEC and highlight our contributions. Section 3 presents the overall system architecture. First, it reviews the overall concepts of the proposed orchestration and network architecture. Second, it illustrates the diagrams of each player and explains the target scenarios. Section 4 elaborates on the proposed implementation of orchestration and edge computing. Section 5 describes our testbeds, outdoor PoC, and the initial measurement results. Finally, Section 6 provides concluding remarks.

## 2. Related Work

This section introduces the related research on the key technologies of MEC in order to highlight their differences from this paper. Table 1 summarizes all related works mentioned in this section.

### 2.1. MEC Architecture

In this paper, the role and location of MEC are both themes. Related research and investigation results are discussed below. In [29], MEC is regarded as one type of edge computing, while architectures such as Edge/Fog Computing are proposed. While there is a concept-based discussion, it does not refer to the players. In the architecture shown by ETSI in [30,31], it is possible to deploy MEC in front of the core functions. In [30], they authors mainly provide an example of use cases where the MEC holder is an existing operator, without specifying MEC location. In [31], the authors discuss the collaboration between multiple legacy operators without discussing other types of players, such as local players or third parties. Although [32] makes architectural proposals focusing on MEC’s NFV capabilities, the authors do not delve into the specific component level. References [33,34,35] are summaries from the viewpoint of MEC functions. In [33], there is a discussion about how to operate DNS on the architecture in order to reduce the connection latency to MEC. There is a discussion focusing on ETSI architecture regarding MEC deployment in [34]. In addition, in [35], an architectural discussion combining Information-Centric Networking (ICN) and MEC is provided. However, the division of responsibilities is missing, because the role of the architecture is not identified. The differences between the above-mentioned works [33,34,35] and this paper involve the viewpoints of each application function (e.g., RAN, Core, MEC) that the owner holds. Most significantly, this paper answers the question of who owns the E2E system functions in MEC architecture.

### 2.2. MEC/Cloud Computing Cooperation

Sharing computing resources with MEC and cloud, in other words, when applications continue to be provided via MEC or cloud deployment without needing to be aware of their physical location, is a key technology. Thus far, various discussions have been held regarding offloading using MEC and cloud [36,37,38,39,40,41]. References [36,37,38] have simulated a basic computational model that divides processing tasks between MEC and cloud to minimize latency or power consumption. A more complicated definition of objective function and analysis considering the queue of computing processing is performed in [39,40,41].

On the other hand, distributed computing has been discussed by providing the MEC architecture in a hierarchical design [42,43,44]. In [42], the authors proposed a hierarchical edge cloud that distributes and deploys computing in order to reduce the amount of traffic on the backhaul side. Further, in [43], the data sent by User Equipment (UE) is first aggregated on MEC, and MEC performs the first-order analysis of the aggregated data. By transferring only the first-order analysis results to the cloud side, a hierarchical edge architecture is adopted to reduce the amount of traffic in the backhaul network and analyze unique data locally. In addition, data collected by MEC/cloud is stored in multiple different layers of information, such as a dynamic map in which essential map information and time-varying data are embedded. An architecture that links mapping information to applications that enables distribution and linkage of data from the cloud to each MEC has been being considered as well [44]. According to the related research mentioned above, the current deployment of applications in MEC or cloud has used the catalog of images. As micro-services that divide application functions have already attracted attention, in Beyond 5G, deploying what is needed at the required location and time will be necessary without being aware of the site (MEC or cloud) for each application function. Therefore, this paper clarifies how to deploy applications in MEC or cloud according to the user’s registration information based on subscription.

### 2.3. MEC Implementation and Verification

Here, we discuss two points with respect to the various implementations of MEC: the orchestrators and the PoC that are distributed worldwide. First, regarding orchestrators, in [45], the MEC Service Function (MSF) is considered an orchestrator. Furthermore, the authors discuss that in MSF, applications are deployed either in MEC or the cloud. In [46], the authors describe signaling to MEC and discuss how users connect with MEC applications. However, based on the above, there is no discussion about who will hold what functions as MEC/Cloud Orchestrator, and this is an important issue to be discussed. Thus, the present paper proposes who should have what functions in the MEC/Cloud Orchestrator.

Next, in [47], performance is evaluated by implementing the edge computing system in a chip set. In [48], the authors implemented a framework linking edge computing and cloud computing as a use case. In addition, other researchers have implemented prototypes for IoT devices [49]. There are examples of implementing a controller for an edge computing platform. For example, [50] implements container-based Network Functions Virtualization Infrastructure (NFVI) control using Kubernetes. References [51,52,53,54,55] discuss the comparison of Fog computing, cloudlet, and MEC. Regarding to application implementation in edge computing, several studies have implemented and demonstrated experiments focusing on Augmented Reality (AR) and image analysis and discussed processing effectiveness at the edge. Based on the above explanation, it is not easy to understand the effect of MEC on PoC because the system has not been implemented in terms of E2E, and thus the effect of MEC on PoC cannot be evalulated by comparing the performances offered by the E2E network without any MEC and offered by the MEC deployment. Therefore, in the present paper, the system is implemented in E2E and deployed outdoors. By evaluating PoC, we validate MEC in E2E and show its effectiveness.

From the above, we now briefly explain the differences between the two points of the edge computing demonstration experiment and the implementation of the orchestrator, respectively. There has been no discussion about the kinds of applications running on virtualization platforms in edge computing regarding the former. Therefore, the effect of edge computing is difficult to understand because it has not been implemented based on the discussed architecture even in demonstration experiments. Regarding the latter, the use case of the orchestrator has not been defined, with only the control of virtualization implemented. Nonetheless, it is a known technology, and there has been discussion about the orchestrator, including Management and Orchestration (MANO). However, many uncertainties remain regarding E2E system implementation and the design of MEC/Cloud Orchestrator including scenarios.

### 2.4. MEC Business Discussion

Research related to trend surveys on service use cases using MEC with players are described in this section. There are several elements to the service use case. For example, it involves efforts and business models, such as establishing a consortium and collaborating with several companies in order to verify new technologies with PoCs and submit/propose a requirement definition to a standardization body such as 3GPP/ETSI/ITU-R. The 5GPPP (5G Infrastructure Public Private Partnership) established by the European Commission, which is the policy body of the European Union, has proposed use cases of MEC and described the advantages of MEC architecture and virtualization [56]. In addition, the open EDGE computing consortium has built the Living Edge Lab as a hands-on project focusing on new technology verification such as application and platform tool verification for Edge and architecture verification [57]. There is a movement to establish an open lab and prepare an environment where an open lab could perform various PoC verification immediately. In the past, it was a flow of conducting desk studies and simulations, design, and PoC. However, as of now, with the variety of tools (e.g., Open Source Software) and prototypes (e.g., Arduino) available today, software can quickly realize new ideas. Therefore, research has become more agile and development can repeatedly develop the prototypes of research ideas for verification [57,58].

In contrast, regarding the business model, several kinds of research accompany the introduction of MEC. For example, ref. [59] deals with the conceptual level, including MEC handover to another MNO by having the MNO network hold MEC. Therefore, the existing operator holds MEC in many architectures [60,61]. However, there is a problem in that these references have many conceptual levels and are not yet mature, because there are many uncertainties with the methods of deploying an application possessed by third parties. Among these, authors have discussed the business model of MEC, proposed a new local/private operator who should hold MEC, and performed numerical analysis by simulation [25,26,27,28]. The present paper discusses who holds MEC/Cloud Orchestrator, which has not been discussed yet. In addition, the author proposes details regarding to the design of each holder based on the content of the discussion.

## 3. System Architecture

This section elaborates the entire architecture and use case concept proposal, including an introduction of each player and a clear description of their roles.

### 3.1. Concept Overview

Figure 2 describes the proposed concept with use cases. In this figure, our target scenario includes use cases such as stadiums, campuses, workplaces, and real estate agencies/post offices with local bases in various places and their own space. The users of MEC purchase subscriptions via the control plane in lower frequency bands that have comprehensive coverage, such as LTE, and acquire computing resources that they can use to obtain several kinds of applications (e.g., mandatory, frequent use). It can be used in a form that suits each scenario, such as cache content and offloading processing. Furthermore, the allocated computing resources can be used locally via the user plane in a closed state to receive the service at a higher data rate and lower latency. Therefore, it is possible to obtain lower-cost service, because it does not use network facilities such as backhaul networks and Internet connection. In addition, applications and data are automatically taken over as one in order to more seamlessly share information or virtual space with others. On the other hand, it is possible to receive local-specific services (e.g., VR attractions, AR autonomous driving support). An MEC/Cloud Orchestrator can control MEC or cloud by receiving and responding to requests from users and determining the resources required to realize the above services.

### 3.2. Architecture

The Private/Local Telecom Operator has an important role in driving local service and providing critical application services. Figure 3 shows the entire system architecture that makes the above concept feasible. In this figure, there is a Private/Local Telecom Operator, a Legacy Telecom Operator, a Cloud Owner, and an MEC/Cloud Orchestrator. Based on the application requirement (e.g., minimization of latency, minimization of CAPEX/OPEX) and the traffic requirements of the end user, the MEC/Cloud Orchestrator can automatically deploy the application for both MEC provided by the private/local operator and cloud resources/platform provided by the Cloud Owner. With the above logic, it is possible to control the traffic generated by the end-user and obtain benefits such as traffic reduction, suppression of network congestion, and high security [62,63]. In addition, from the end user’s perspective, the application throughput in MEC is faster than during regular use on the cloud, which improves QoS/QoE. The network shared with end users must be recognized as a personal space network (individual slicing of shared resources). Here, the difference between the proposed architecture and MEC provided by the conventional telecommunications operators (Legacy Telecom Operator) is that it is a local business that holds physical resources (locations) where edge computing such as MEC can operate closer to the user while offering free utilization space for Commercial Off-The-Shelf (COTS) servers. Furthermore, the Private/Local Telecom Operator provides end user data and supports control data via the Legacy Telecom Operator. In other words, the Legacy Telecom Operator provides a stable RAN service from the perspective of coverage, and the Private/Local Telecom Operator provides the Application service for hotspots and local points. Therefore, the difference is that the network is closed to end users in terms of local secure data and assurance of latency.

Figure 4 shows a network configuration diagram of the Private/Local Telecom Operator network in the above architecture. This figure focuses only on the Private/Local Telecom Operator network. First, installation of a Radio Unit (RU) can support higher-frequency bands (e.g., Sub6/mmWave) with a local RAN network in order to accommodate as much traffic as user plane data in a hotspot. XR/AR/VR/UAV, C-V2X, Robotics, IoT, etc., are examples of terminal devices that connect to the control/user plane. The RU is connected to the virtualized Distributed Unit (DU) pool via the fronthaul. The vDU is connected to the virtualized Centralized Unit (CU) pool via the F1 interface in the midhaul. Here, vCUs are classified into vCU-CP with a control plane and vCU-UP with a user plane, and they are deployed in the same/different place, respectively. Packets GTP encapsulated by CU-UP are GTP decapsulated in MECs that hold the UPF function. MEC analyzes the destination information in the packet and refers to the registry information and whether it is operating as MEC Apps. If there is a target MEC Apps, the traffic is passed to MEC Apps at the TCP/IP layer; if not, it is encapsulated again by GTP, and the traffic is passed to the mobile network again. Meanwhile, vCU-CP requests the X2 interface in 4G BBU to receive RU support for a wide area. Here, as an example, we request a 4G BBU to anchor via the X2 interface. As a result, terminal devices can use C-Plane and U-Plane, setup Protocol Data Unit (PDU) sessions, and receive services.

### 3.3. Strategy of Each Player

Each player’s role and relationship is discussed in this sub-section. Figure 5 shows the relationship chart of six possible players. The details are described in Section 3.3.1–Section 3.3.6.

#### 3.3.1. Private/Local Telecom Operator

The Private/Local Telecom Operator mainly provides two services: an end-user application service and a business-to-business service. Regarding the former service, there are generally multiple application service types from a third-party perspective. The service provider purchases the application itself from a third party; advertising is done in the application, as is billing [64,65]. Most application providers use Freemium and subscription models [66,67]. If these models are applied to an MEC-oriented platform, the Private/Local Operator has two types of service options. First, it can collect the cost of using the application itself from the end user based on the resources running at the MEC. Second, it can license the application.

In the latter case, the MEC environment must have platform correlation in order to run applications running in the cloud. Correlation allows MEC/Cloud Orchestrators to deploy applications according to conditions by providing support for interfaces such as representational state transfer (REST) for higher levels. For that purpose, the MEC/Cloud Orchestrator may wish for a consultant to have Cloud Owner build a correlated platform, requiring a business alliance scheme.

#### 3.3.2. Legacy Telecom Operator

The existing Legacy Telecom Operator has been providing mobile communication services to end users using its infrastructure equipment and spectrum resources assigned by the government. However, in the B5G era, they will not always be able to survive due to the exhausting spectrum resources, part of which are released for regional operators, e.g., private 4G/5G and MVNOs. Therefore, it is necessary to provide new operators with development assets and infrastructure such as equipment and RAN/Core software to expand their service areas. Legacy carriers need mobile infrastructure such as RAN and Core using dedicated servers, while the development/innovation of RAN using virtualization technology through NFV and open interfaces empowered by O-RAN alliance supports the above framework. As described above, various technologies and existing infrastructure (mobile backhaul, macro-coverage, etc.) can be provided to private/local carriers. Therefore, by anticipating the technological background and future growth, it is possible to provide support from the viewpoint of operation.

#### 3.3.3. Vendor Supplier

The vendor suppliers mainly provide hardware such as RU and COTS servers, which are general purpose servers. In addition, it is possible to provide a their virtualization platform software and network functions (RAN/Core software, MEC platform) based on standardization (e.g., ITU-R/3GPP/ETSI/O-RAN). As standardization organizations play a central role in implementing multi-vendor support to avoid market monopoly by one vendor’s specifications, each software must be in line with the standardization specifications, and the Private/Local Telecom Operator must be multi-vendor. On the other hand, in order for the Private/Local Telecom Operator to develop the above products, it is necessary to acquire human technical resources, as many specific skill-holders are required. Although it is difficult to describe the above option, there is a model, provided by the Legacy Telecom Operator, for all resources, making it possible to obtain the needed skill sets for both equipment provision and operation.

#### 3.3.4. Cloud Owner

Cloud Owners can provide cloud resources (e.g., computing, network, storage). Each resource can manage the application-like cycle management with the officially released interface (e.g., Restful API, CLI, etc.). In addition, when using the Orchestrator held by the Cloud Owner, it is possible to run the application using unofficial information (physical server/network location, etcetera). On the other hand, from an application perspective, applications need compatibility support (e.g., without hard cording, container/virtual machine, support north-bound/south-bound interface) to deploy on both platforms of MEC/Cloud. The virtualization platform requires that necessary conditions such as the driver of the virtual interface and the number of virtual interfaces will occur for each OS and application to ensure compatibility between MEC and cloud. Furthermore, it is necessary to provide a virtualization platform that can support virtual machine-based and container-based functionality at the same time. Finally, it is necessary to create rules for each holder, such as cluster-based and server-based.

#### 3.3.5. Third Party Application

A third party is required to design and create a microservice architecture model in consideration of deployment cases for the application itself and each function level in MEC/cloud. Developed software functions can be deployed on a virtual machine or container basis, and this functional split should be optimized. In addition, the development of new content and the provision of patches plays a role in expanding application support. Use case examination is needed in collaboration with the Private/Local Telecom Operator or Cloud Owner in order to satisfy the requirements for application functions. Meanwhile, the Third Party independently registers for a subscription, which requires examination/inspection by the application platform owner, e.g., Apple Store/Google Play. Therefore, the application development process needs to take into account existing business models such as the application-only purchase model, function purchase model, advertising revenue model, free model, subscription model, and donation model [64,65,66,67].

#### 3.3.6. MEC/Cloud Orchestrator

The MEC/Cloud Orchestrator needs to consider two optional model use cases: centralized and decentralized. In the centralized type, the Orchestrator supports multiple Private/Local Telecom Operators and is held by the Cloud Owner. On the other hand, functions may exist on the Internet in the distributed type, while other parts may be retained by each Private/Local Telecom Operator. For the centralized type, it is possible to deploy an application to MEC that considers the end user usage information (e.g., usage access log) on the cloud. For the distributed type, it is possible to deploy an application to MEC that guarantees high security. The functions provided as Orchestrator manage the catalog image of the application and support the northbound and southbound interfaces (MEC platform, cloud platform, DNS entry, user request management), with the main function being application life cycle management.

On the other hand, various studies have been conducted from the user point of view regarding MEC and locality. Because there are many unstudied factors regarding how to control MEC, this paper provides a detailed explanation of the MEC/Cloud Orchestrator. In addition, we explain the implementation of the MEC/Cloud Orchestrator in the next section.

## 4. Implementation of Proposed Architecture

This section describes the implementation of the MEC/Cloud Orchestrator for deploying applications on MEC or the cloud. The Orchestrator plays an essential role in acquiring end-user requests and deploying the application based on the collected information and algorithm. The architecture and implementation sequence are included separately for the centralized and distributed types explained in the previous section.

### 4.1. Function-Level Description of MEC/Cloud Orchestrator

Each of the functions of the GUI and the Orchestrator is explained below in detail. The GUI function consists of the GUI View function and the Subscription Management function. The Orchestrator function consists of six parts: the Service Query function, Service Registry function, Database Update function, Image Registry function, Deploy Judgments function, and Setup Execution function. The GUI View function allows end users to connect to a wide-area wireless network provided by the Legacy Telecom Operator via the control plane using an HTTP/HTTPS. After accessing the GUI, the end-user begins to use it by registering a subscription (billing) with a secure connection. Here, the user management access information and subscription management used by the end user at the time of access by the GUI function are performed by the Subscription Management function. In addition, the application subscription information is managed by the Subscription Management function. On the other hand, the GUI View function functions as a request received on the Web and a transaction role for each part. Requested information for application requirement requests received by the GUI View function are sent to the Orchestrator function via Southbound Interface (e.g., Restful API, Curl, SSH, CLI). Then, the end user can request application deployment according to subscription payment. Meanwhile, the Orchestrator function regularly monitors the resource usage/allocation of cloud and MEC for computing/network/storage and the Database Update function updates/manages on Database based on monitoring info. The Service Registry function registers application deployments requested by the Service Query function. The application selected here starts the deployment process using either the one registered in the Image Registry function provided by the Third Party in advance, or the cached content generated by the learning function by AI/ML. When starting the deployment process, the Deploy Judgments function is determined by logic based on the scenario in which either cloud or MEC (e.g., the physical location) possessed by the Private/Local Telecom Operator is registered. Examples of registered strategies include minimum latency, cost, and processing performance. Finally, the Setup Execution function executes the Config set in the virtual machine or container remotely when the deployment is completed. As a result, the end user can receive the service from the deployed application via the user plane.

### 4.2. Centralized MEC/Cloud Orchestrator

#### 4.2.1. Logical Implementation

First, we propose a centralized MEC/Cloud Orchestrator on the Cloud Owner that collectively manages multiple Private/Local Telecom Operators’ MEC and integrates with cloud services. Figure 6 shows a proposal overview. The MEC/Cloud Orchestrator consists of GUI and Orchestrator functions and has northbound/southbound interfaces. For the northbound interface, GUI functions are provided to the end user. For the southbound interface, the Orchestration function is used for application deployment, setup, information collection/monitoring to each cloud, and multiple MECs. Here, managing multiple MECs collectively has three main advantages:(1)The usage log information of the application that the end user has used in the cloud can be used as input information to AI/ML. Then, the cached content can be deployed by AI/ML on MEC as a usage prediction or made known to end users as recommendation information and selected.(2)When the end user has contracts with multiple Private/Local Telecom Operators, it is possible to track the movement of the application used by MEC when the end user moves, because multiple MECs are managed collectively.(3)Because the MEC/Cloud Orchestrator monitors each resource, the awareness of the physical location means that visualization management of the entire network can be performed, allowing network route change in the event of a disaster, etc., to be taken into consideration.

Thanks to the above-mentioned advantages, the proposed centralized scheme can collectively manage and operate multiple MECs.

#### 4.2.2. Sequence Implementation

There are two main steps for the end-user to obtain the Private/Local Telecom Operator’s services. The first is subscription registration, and the other is application deployment.

The implementation detail of the proposed centralized sequences is shown in Figure 7. First, the end user needs to access the GUI view using a registered account. Here, if the user obtains access with a registered account, they receive a success message in GUI view. However, if the user accesses with an unregistered account, they receive a failure message response. The above sequence is a basic rule for blocking unauthorized access using a registered account in Figure 7a. On the other hand, the sequence in Figure 7a acquires information on the cloud and multiple MECs regularly, which is a different sequence from user access. An authentication session can be established because the key authentication format is distributed in advance. As monitored by the polling method, the status of the physical/logical information of the resource area (computing, network, storage) secured in advance on the cloud system is confirmed. Similarly, the status of the physical/logical information of the resource area that can be used as MEC held by each Private/Local Telecom Operator is checked. The database inside the Orchestrator function is updated based on the information learned here. Although Figure 7a exemplifies the polling method, of course, a notification format in Syslog/Trap format is conceivable. Regarding the deployment resource space, as it is assumed that the Cloud Owner holds MEC/Cloud Orchestrator, the application deployment computing resource in the cloud can be changed by the MEC/Cloud Orchestrator according to the usage status of the application. The resource area in MEC can be determined/changed according to the contract with the Private/Local Telecom Operator.

Next, the end user describes the sequence of registering a subscription based on the access information. The end-user makes a subscription registration request. Here, the end user needs to input the Private/Local Telecom Operator service data and select the range of use and the type of service. The Orchestrator GUI then receives a registration request with the GUI function and makes a service registration request to the Orchestrator function. The Orchestrator function confirms whether the resource area on the MEC side is open for the received request. If there is no problem, it provides a successful notification. Otherwise, if the resource area is insufficient, a request is made to the Private/Local Telecom Operator to update the contract in order to obtain the additional resources. The MEC side is notified of the possibility that the resource area is insufficient. The service requirements (latency, cost, etc.) are not positively required. The Orchestrator notifies the end user that the processing performance is not guaranteed in this case. When further subscribing, it is necessary to consider the priority compared to other end users.

Second, the application deployment sequence is shown in Figure 7b. The end-user starts the application deployment based on their subscription. There are three designs as an application deployment method, as follows: (1) specific application deployment method; (2) cloud cached content of application specified in the cloud; and (3) computing resource reservation method.

The deployment method in (1) involves deploying a basic application on a virtualization platform through VIM (Virtualized Infrastructure Management). Method (2) is an intelligent method that generates cache content based on the access usage log information of cloud applications that end users have used thus far, and deploys it on MEC. This method is possible because the Cloud Owner holds the MEC/Cloud Orchestrator. Regarding (3), unlike (1) and (2), it is a method of reserving computing resources in advance. The resource reserved here could be used as a calculation resource or could be used as in (1) and (2). Subscriptions only occur after use begins. With the above, it is possible to deploy the application on the cloud or MEC.

### 4.3. Distributed MEC/Cloud Orchestrator

#### 4.3.1. Logical Overview

In a distributed manner, the orchestrator function and GUI function are separately deployed to each Private/Local Telecom Operator and the Internet, respectively. An overview of this architecture is illustrated in Figure 8. The GUI functionality of the MEC/Cloud Orchestrator is deployed at a higher level, e.g., on the Internet, allowing end users to access it.

Meanwhile, a part of the orchestration function, that is, the deployment function, is assigned to each Private/Local Telecom Operator.

The GUI function determines which Private/Local Telecom Operator’s service range is provided based on the end user’s registration information and requests it via the internal management interface. Based on the requested information, the Orchestrator function of the Private/Local Telecom Operator receives the information at the API interface published by the Cloud Owner via the northbound interface. This is compared with MEC information acquired via the southbound interface and then the deployment destination is registered. Finally, the application is deployed based on the scenario’s judgment logic. There are a plurality of merits when an MEC/Cloud Orchestrator is used. The distributed manner has several advantages when an MEC/Cloud Orchestrator is employed and managed for each Private/Local Telecom Operator, as listed as below:(1)It supports the concealment of confidential information such as network information, physical context information, and server information, for each Private/Local Telecom Operator.(2)It can make recommendations to end users using the Private/Local service according to predicted regional information by algorithmizing the application in MEC which they hold as input information to AI/ML.(3)When a Private/Local Telecom Operator covers multiple areas, the log/tracking data and updating of applications used by end users can be shared among regions.

Because of the advantages described above, a distributed architecture can be a solution for managing and utilizing MEC for each Private/Local Telecom Operator.

#### 4.3.2. Sequence Implementation

The basic sequence flow triggered by the end user is the same as in Section 4.2.2. Here, the supplementary information in the distributed-type sequence is explained with refeence to Figure 9.

In Figure 9a, regarding the sequence of regularly collecting data on the cloud and MEC, the cloud side needs to obtain the physical/logical data via the interface officially published by the Cloud Owner. On the other hand, because the resource area on the MEC side is inside the network, the resource controller considers information such as RAN. MEC then becomes available because the main function of the MEC/Cloud Orchestrator is under the Private/Local Telecom Operator. Thanks to the above information, a roadmap for future hardware procurement, etc., could be planned. The Private/Local Telecom Operator can easily decide on the purchase order supported by the MEC/Cloud Orchestrator. Regarding subscription registration, when resource usage is about to exceed the capacity pre-reserved on the cloud, the MEC/Cloud Orchestrator can request the cloud owner or end user to acquire additional resources.

Next, the application deployment sequence in Figure 9b is described below. The application deployment sequence is almost the same as in the centralized type. The difference is the deployment method, which uses the localized cached contents method. In the localized cached contents method, the subscription of registered users is based on the analysis result of the usage logs of the ranking information in which many applications are used, and the stored end user information is based on the history information of the application used by the Private/Local Telecom Operator. In the localized cache contents method, the cached application is deployed based on the analysis result of application ranking and trend provided by the end user. This kind of application deployment method can be classified as localized information-based digital twins.

## 5. Performance Evaluation of MEC B5G Cellular Networks

In this paper, outdoor PoC work was conducted with realistic 5G. The detailed scenario of this field trial is presented and the benefits of Private/Local Telecom Operators are shown by the results.

### 5.1. Proof-of-Concept Description

The experiment field structure is shown in Figure 10. The outdoor PoC field was constructed at the Tokyo Institute of Technology, Ookayama campus. As the 5G environment, multiple RUs were deployed for n77/n257 frequency bands and one LTE sector was deployed for various kind of applications. Table 2 shows hardware equipment and radio information, such as the frequency of each RU. 4G uses 5 MHz channel bandwidth (UL: 1825–1830 MHz, DL: 1730–1735 MHz) in Band 3 (FDD), while 5G bands are prepared in both Sub6 n77 (TDD, 3.8–3.9 GHz) and mmWave n257 (TDD, 27.0–27.4 GHz). In the case of the mmWave band, 400 MHz Bandwidth, i.e., 4CC (4 × 100 MHz Component Carrier), was assigned to the downlink, while that for the uplink was 100 MHz Bandwidth, i.e., 1CC (1 × 100 MHz CC). As a network connection, Edge Cloud was defined from after Fronthaul to before the Internet Security Getaway. 4G/5G of RAN (vDU/vCU) was deployed in the Edge cloud, and MEC and Apps that could be capsulized by GTP of User Plane were prepared. In addition, a cloud vApps server was prepared on the cloud side. In the vApps, iperf3 software was installed to compare the communication performance of MEC and cloud. As a user equipment terminal, a 5G-compliant Rakuten Big s [68] was used. Its local App can specify IP for Iperf3 of vApp in MEC and register itself in DNS for IPerf3 of Apps in cloud. Each of those settings can be confirmed by connecting with the specified Name. In addition, 4G was used for C-Plane signalling, and measurement was performed based on conditions in a state where PDU session establishment was completed. In other words, PoC measurement starts from UE connection establishment, and does not take into account the latency imposed by the attach procedure or any handover scenario. This is why only a stational terminal was involved in the PoC and the measurement was performed in a static condition, as the main purpose of this PoC was to verify how essential and efficient the use of 5G MEC is for Private/Local Operators scenarios from the user plane performance standpoint. This PoC field coverage could be expanded by additionally deploying more RUs to fully support the 5G area. Regarding the connections of the terminal, a PC was connected to the terminal via USB3.0 and the log information of the terminal was collected via XCAL [69]. With this measurement tool, it is possible to acquire information such as RSRP, each protocol sequence, physical/MAC layer information, and wireless throughput. The captured data can be combined and analyzed using XCAP [70]. Performance tendencies with different network configuration for fronthaul/backhaul will be investigated separately. For measurement, the application deployment location (either MEC or Cloud) is determined by the objective function in order to minimize the cost that satisfies the latency conditions.

End users would like to select the more inexpensive computation environments. Here, the cost model based on [28] was calculated by numerical analysis conducted with the actual field measurement results. The actual results in the testbed field were obtained by measuring the performance on the RAN side using XCAL.

Meanwhile, the logic of judgment needs to be added in the MEC/Cloud Orchestrator, as shown in Figure 3. In this paper, we explain how to determine the minimum latency. The latency on the MEC side can be determined by the computing process, because it is assumed that the MEC will be deployed on the RAN side. Regarding to the latency imposed on the cloud side, it is necessary to consider two kinds, namely, backhaul network bandwidth and computing processing. The cloud side needs to evaluate both the latency effect and cost increase, whether the network bandwidth or computing resource is added. The MEC latency $t_{k,j}$ [28] is expressed as
(1)$$t_{k,j}=\frac{\alpha {}_{k}w_{k}}{N_{\mathrm{MEC}}f_{k,j}}+\varepsilon_{j},w_{k}=\delta b_{k}$$
where $N_{\mathrm{MEC}}$ denotes the number of MEC resources decided by the private (local) telecom operator’s strategy, $w_{k}$ (CPU cycles) represents the task converted from information $b_{k}$ with task weight δ, computation resource is expressed as $f_{k,j}$, and $\varepsilon_{j}$ is the processing queue on the MEC side as seen in Equation (Equation 1). On the cloud side, the latency $t_{k,\mathrm{cl}}$ and cost $c_{\mathrm{cl}}$ are expressed as,
(2)$$t_{k,\mathrm{cl}}=\frac{\left(1-\alpha {}_{k}\right)b_{k}}{N_{\mathrm{BH}}}+\frac{\left(1-\alpha {}_{k}\right)w_{k}}{N_{\mathrm{cl}}f_{\mathrm{cl}}}+\varepsilon_{\mathrm{cl}}$$
where $N_{\mathrm{BH}}$ denotes backhaul capacity, $N_{\mathrm{cl}}$ is the number of cloud resources, $f_{\mathrm{cl}}$ is computing resources, and $\varepsilon_{\mathrm{cl}}$ denotes the processing queue in the cloud, as seen in Equation (Equation 2). Because the measurement is performed using one sector, sector *j* is set to 1 and user *k* is one UE, because RU is able to allocate the maximum resource block (RB). It is possible to input highly reliable information as an input in the numerical calculation based on the measured delay performance.

### 5.2. Edge Platform Virtualization Implementation

The edge computing platform held by the Private/Local Telecom Operator requires that various applications (RAN/Core/Apps) run on shared hardware. Figure 11 shows the architecture that the applications can run on the virtualization platform. In this figure, COTS HW requires a Management (MGMT) GUI function that remotely controls the OS and power supply. For the OS, a general-purpose OS such as Ubuntu/CentOS/RedHat/Windows is used. As a Virtualization Platform, HW has vFPGA/vMEM/vCPU/vGPU/vSSD/vHDD/vNetwork/vNIC/vDriver that can be provided as virtualization as NFVI and can be managed by VIM, and specify it at the time of deployment according to the application requirements for virtualization. An API (Restful/CLI/SSH/YANG) can control VIM for NFVI via MGMT NW. RAN Software/Core Software/MEC Platform/MEC Apps can be operated as an application. RAN Software is divided into vDU/vCU. Furthermore, vCU is separated by CUPS and is divided into vCU-CP/vCU-UP. The interface between vDU and vCU is divided by Split Function (e.g., Split Option 6). Core Software is a 5GC-based function that can be deployed as needed, such as UPF, AMF, and SMF. MEC Platform and MEC App, etc., mainly indicate applications used by end users when a local breakout is required. As it runs on the same COTS HW as the above application, it can be operated by sharing resources and managing and monitoring it on the virtualization platform. In addition, the virtualization platform is adopted as a single factor without multiplication in order to accommodate virtual applications for high processing performance.

### 5.3. Result of Field Trial

The basic throughput performance of Sub6 and mmWave was measured through the NSA configuration. The measurement results in E2E were approximately 0.9 Gbps for Sub6 and 1.6 Gbps for mmWave under the same 4G coverage in Table 3. This result includes throughput on the 4G side and is based on RF level, which depends on the development status of RAN software. It should be noted that the throughput in this measurement could not be the maximum performance. Meanwhile, in order to highlight the benefit of MEC, which supports stable communication compared to the internet, we executed the throughput performance from Ookla speedtest [71], which works in San Francisco, United State, via mmW from UE. As shown in Table 3, the performance in MEC via mmW is higher than on the internet. Hence, the performance of MEC is not only more stable, it is higher than the internet.

As for the latency performance, one packet of 56-bit data was transmitted via 5G; thus, it does not occupy the RB and compares to throughput, allowing a quantitative evaluation to be performed. Figure 12 shows the latency result between MEC and the internet. For the internet servers, the comparison of MEC was conducted using officially prepared ping servers in Hokkaido, Japan [72] and San Francisco, United States. For the internet, the former was executed by ping from UE with an IPv4 address in a Hokkaido ping server, while the latter was executed via Ookla speedtest [71] in UE.

In this figure, Internet (1) refers to the Hokkaido Ping server (ping-hokkaido.sinet.ad.jp) and Internet (2) to the San Francisco, United States server. These results show the average performance of ten measurement trials in a 100 s period at UE. From the figure, it can be seen that the average latency of MEC via Sub6/mmW is approximately 33 ms, which is an improvement of about 20 ms compared to Internet (1). In addition, it shows that the improvement in latency depends on the server’s location on the internet based on physical distance. Based on the above results, we can estimate that the latency between the RU and the CU was about 5 ms when installed on the near side of the base station, meaning that the latency can be reduced to about 5 ms to 15 ms. These results were implemented assuming a Private/Local Operator as an MEC that partially verifies the designed MEC/Cloud Orchestrator. As a result, the application was correctly deployed and implemented in E2E, and the communication stability and delay were more stable and lower than in the case of the internet cloud. In other words, the throughput in MEC is more stable than without MEC, because the distance of the E2E network is shortened and the network’s condition becomes more stable as there are fewer in-between network devices.

## 6. Conclusions

This paper addresses the challenge of managing and deploying applications in MEC. In particular, we propose a detailed implementation for an MEC/Cloud Orchestrator. In addition, we design a case where the MEC/Cloud Orchestrator is owned by either an existing Cloud Owner or a Private/Local Operator that splits the functionality of the MEC/Cloud Orchestrator. During the design process, it was found that deploying the RAN and MEC on a virtualized platform and renting other equipment from Legacy Telecom Operators facilitates the participation of Private/Local Operators in the MEC ecosystem. For system implementation of this proposal, we designed an E2E system and constructed a PoC field at Ookayama Campus of the Tokyo Institute of Technology. Using this field, an edge cloud was implemented on campus and SOTA hardware and software were deployed. We conducted a PoC through an outdoor field trial to verify the essential effectiveness of the proposed concept from an E2E perspective. The obtained verification results showed lower latency and more stable communication than current cloud services. In the future, in order to generalize the implementation system, we intend to consider and implement open source-based system implementation and microservices. In addition, we intend to utilize the PoC outdoor field constructed in this paper in a variety of Beyond 5G research.

## Figures and Tables

**Figure 1 sensors-22-05145-f001:**
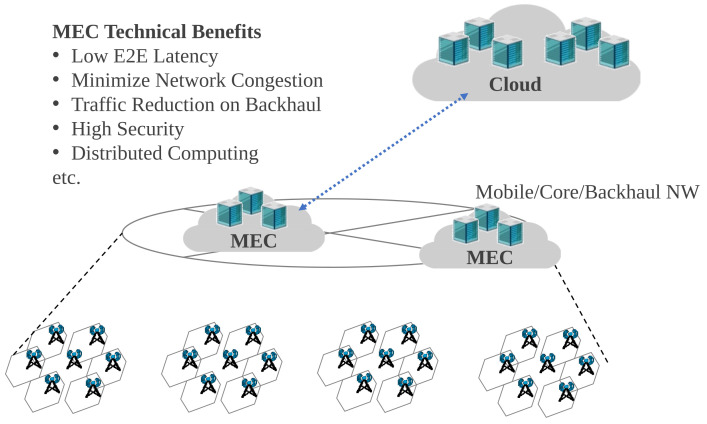
Benefits of MEC compared to cloud systems.

**Figure 2 sensors-22-05145-f002:**
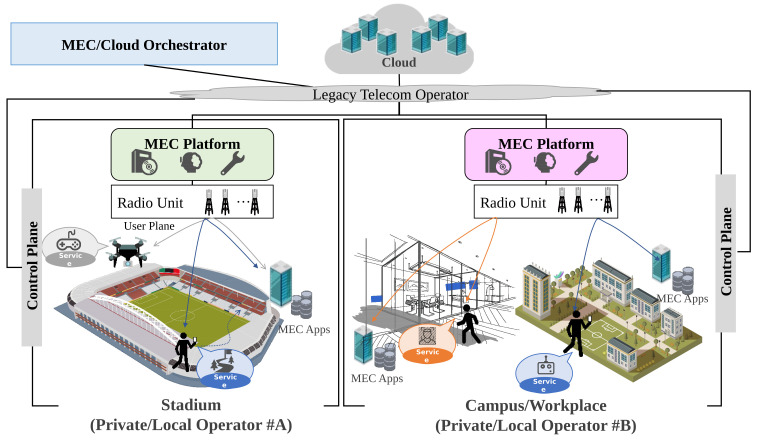
Overview of concept proposal for Private/Local Telecom Operator.

**Figure 3 sensors-22-05145-f003:**
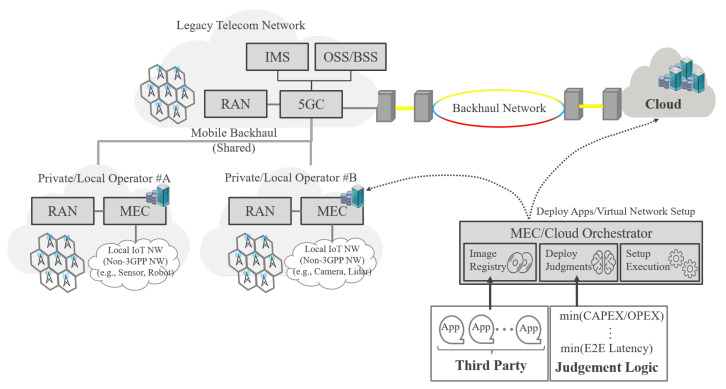
Illustration of the system architecture of an MEC/Cloud Orchestrator for a Private/Local Telecom Operator and Cloud Co-operation.

**Figure 4 sensors-22-05145-f004:**
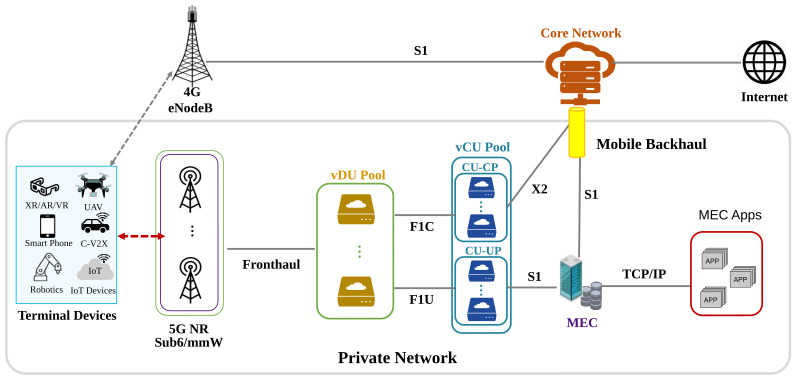
Network Architecture of Private/Local Telecom Operator.

**Figure 5 sensors-22-05145-f005:**
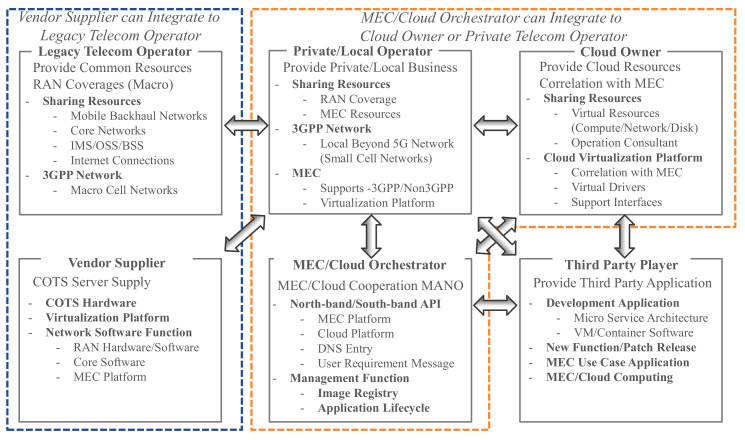
Relationship chart for each player.

**Figure 6 sensors-22-05145-f006:**
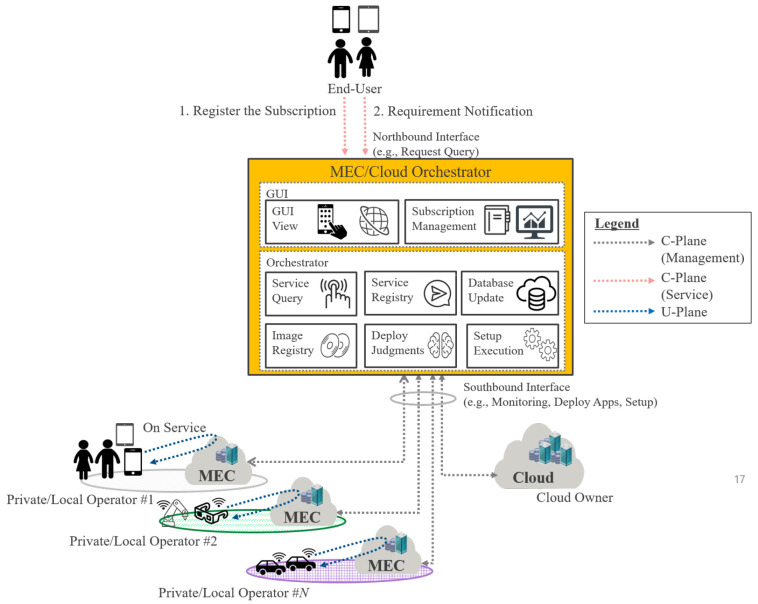
Illustration of the Centralized Type of an MEC/Cloud Orchestrator.

**Figure 7 sensors-22-05145-f007:**
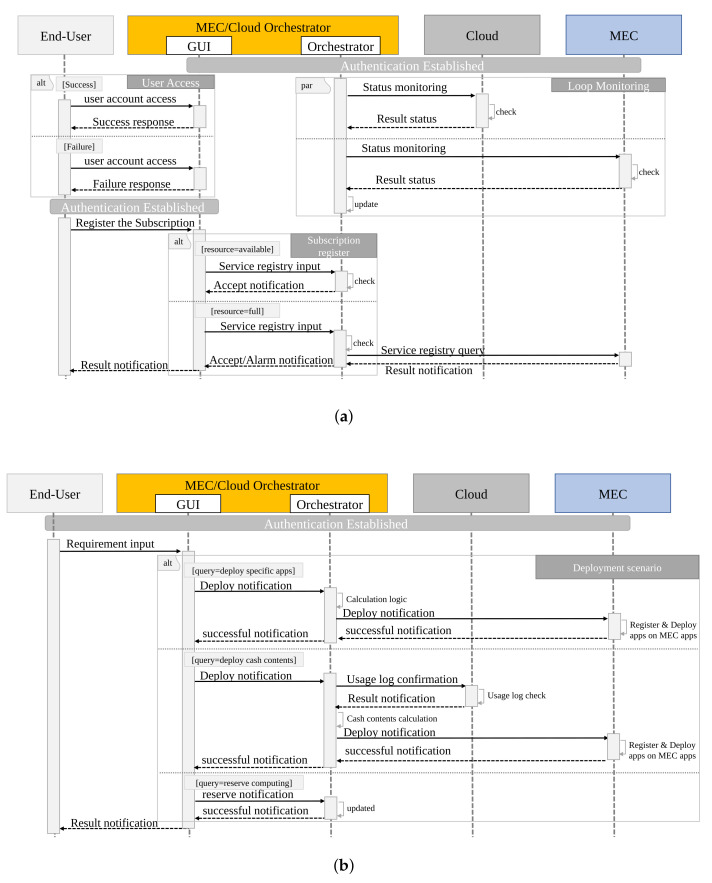
Sequence of MEC/Cloud Orchestrator for implementation: (**a**) subscription registration process and (**b**) deployment process.

**Figure 8 sensors-22-05145-f008:**
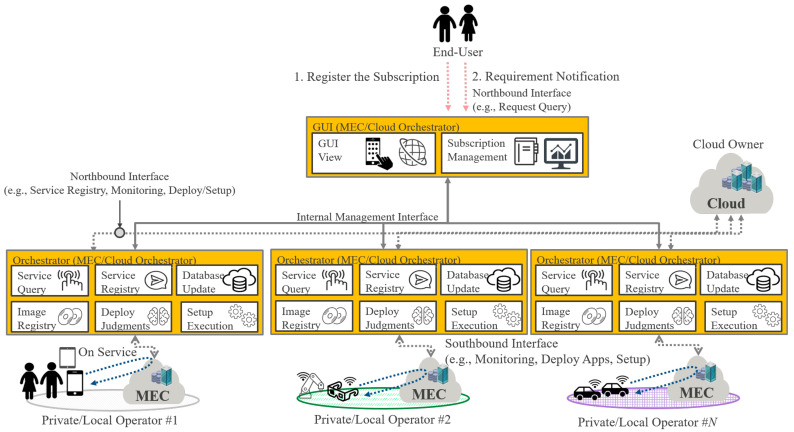
Illustration of the Distributed Type of an MEC/Cloud Orchestrator.

**Figure 9 sensors-22-05145-f009:**
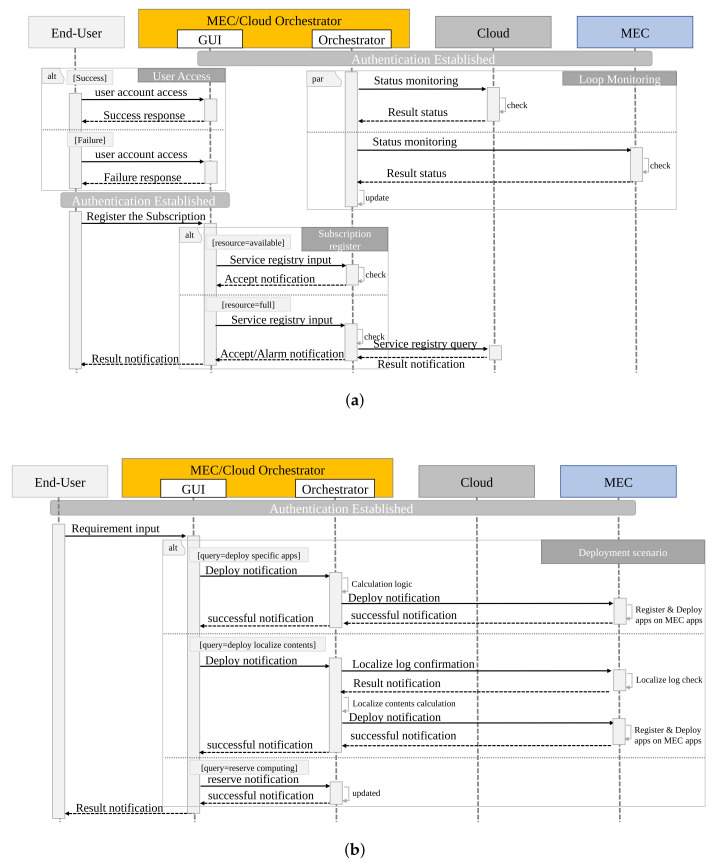
Sequence of MEC/Cloud Orchestrator of distributed type: (**a**) subscription registration process and (**b**) deployment process.

**Figure 10 sensors-22-05145-f010:**
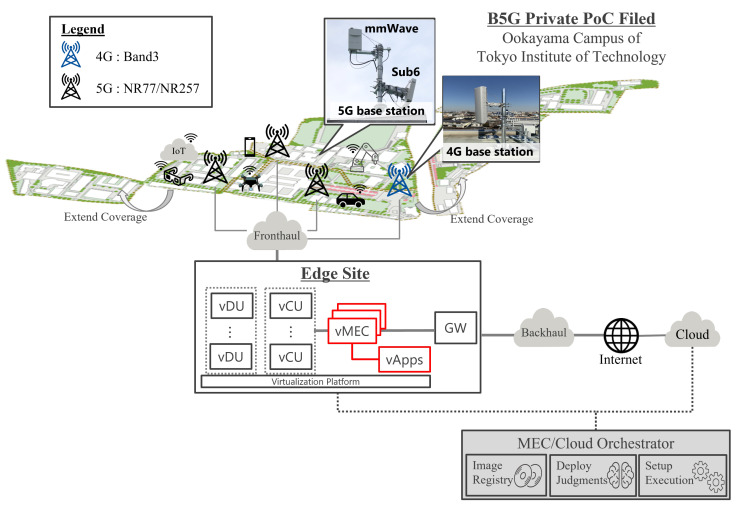
Outdoor PoC Field Design.

**Figure 11 sensors-22-05145-f011:**
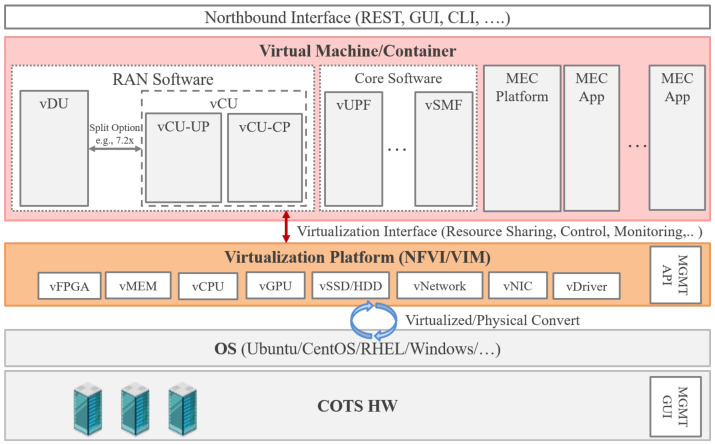
Edge Platform.

**Figure 12 sensors-22-05145-f012:**
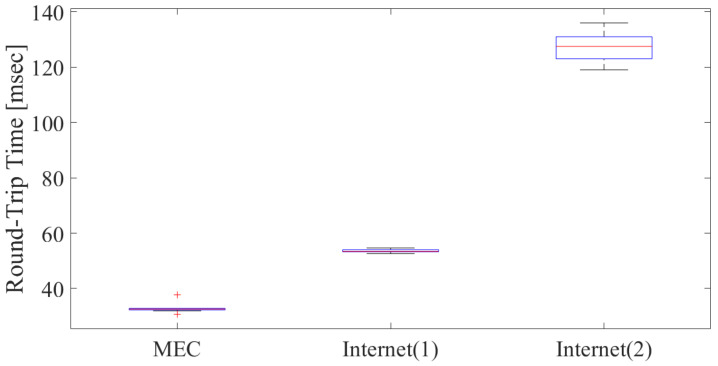
Latency comparison of MEC and internet.

**Table 1 sensors-22-05145-t001:** Comparison of Related Technical Works from Several Perspectives.

Aspect	Ref	Main Contribution
MEC Architecture	[29]	Edge/Fog Computing Proposal Concept-Based Architecture.
	[30,31,32]	MEC deployment scenario in front of Core networks; assumed that only player is Legacy Telecom Operators.
MEC/Cloud Computing Cooperation	[33,34,35]	Viewpoint from function-level architecture, such as DNS, Information-Centric Networking, and ETSI-based.
	[36,37,38,39,40,41]	Several works have already proposed offloading cooperation such as latency and power consumption with several architecture models.
	[42,43,44]	Optimization of telecom operator’s revenue with the number of MEC as well as backhaul owner’s revenue with the backhaul capacity.
MEC Implementation, Verification	[45,46]	Describes the MEC orchestrator and signaling for service provision.
	[47,48]	Demonstration of edge computing: Distributed edge computing, Edge/cloud Cooperation Framework, etc.
	[49,50,51,52]	Platform controller implementation; discussion of implementation comparison of Fog Computing/cloudlet/MEC.
	[53,54,55]	Application implementation (e.g., AR, Image Analysis) in edge computing.
MEC Business	[56,57,58]	Several Consortiums and established Open Labs are discussed with respect to business model.
	[25,26,27,28,59]	Legacy Telecom Operator scenarios in MEC are discussed, along with new private operators.

**Table 2 sensors-22-05145-t002:** Hardware Equipment Condition.

Hardware Name	Specifications
LTE RU	Frequency Band: Band 3 System bandwidth: 5 MHz
Sub6 RU	Frequency Band: n 77 System bandwidth: 100 MHz
mmWave RU	Frequency Band: n 257 System bandwidth: 400 MHz
UE Device [68]	CPU: Qualcomm® Snapdragon™ 765G 5G mobile platform OS: Android Support Band: Band 3/n77/n257 Band3 Tx Rate: UL ≤ ≤ 100 Mbps n77 Tx Rate: UL ≤ 217 Mbps, DL ≤ 2.13 Gbps n257 Tx Rate: UL ≤ 273 Mbps, DL ≤ 2.80 Gbps
PC/Laptop	Model: dynabook G83/DN OS: Microsoft Windows 10 Pro CPU(Phy)/MEM: 4 Core/8 GB USB ports: 2

**Table 3 sensors-22-05145-t003:** Throughput performance in PoC Field employing MEC.

	w/MEC		w/o MEC
	Sub6 n77	mmW n257	Internet
Throughput [Gbps]	0.9	1.6	0.9

## Data Availability

Not applicable.

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
