# Peer review of "MEC/Cloud Orchestrator to Facilitate Private/Local Beyond 5G with MEC and Proof-of-Concept Implementation"

_sensors, 2022, doi:10.3390/s22145145_

Round 1
Reviewer 1 Report
In this paper, the authors have demonstrated proof-of-concept scenarios in 5G and beyond networks with a proposal “MEC/Cloud Orchestrator to facilitate Private/Local Beyond 5G with MEC and Proof-of-Concept Implementation”. The results have been presented in the well-known conferences (preliminary). I have the following recommendation regarding the improvement of the paper.
1. Please fix the typos such as in line#34, a space is required between the “progression and [5], operators and [6,7]—backhaul in line#46, etc.
2. Acronyms need to be illustrated on their first appearance in the manuscript e.g., RAN in line#30 and in the abstract.
3. In line#117, you have mentioned the above-related works. Please mention or cite the paper i.e., is it [33-35]?
4. The paper lacks a comparison with other existing schemes. How can we assure that the proposed methodology is effective in terms of throughput or latency.?
5. In the literature, mention the significance of SDN/NFV as stated in the paper "QoS improvement with an optimum controller selection for software-defined networks"
6. The paper needs a minor proofread
Author Response
The authors’ reply has been included in the attached reply letter.

Reviewer 2 Report
This paper is a significant original contribution of the authors to the 5G communication research area, proposing and successfully implementing a detailed MEC/Cloud Orchestrator providing a PoC implementation of the E2E system.
The efficiency of the proposed solution was practically proved by designing an E2E system and constructing a PoC field at the Ookayama Campus of the Tokyo Institute of Technology. The results show that the average latency of MEC via Sub6/mmW is about 20 msec less than the usual Internet connection.
The research was well conducted, the paper well structured, and the presentation is sound.
I recommend this paper for publication.
Author Response
Thank you for reviewing my paper and giving positive comments.
We tried to do our best for the paper modification based on other reviewers’ comments.
Reviewer 3 Report
Dear Authors,
After reading your paper, I came to the following conclusions about some of the issues you should address:
-
English language and style issues - Grammarly (https://app.grammarly.com) on default settings detected only for the text block resulting from the concatenation of Title+Abstract+Keywords 9 critical alerts (correctness issues) and 12 more advanced ones, namely: Passive voice misuse (6 issues), Word choice (2), Unclear sentences (2), Wordy sentences (1), and Faulty tense sequence (1). This meant a total score of 86 out of a maximum of 100 for this sample above which is not bad at all. However, since you do not appear to be native English speakers, I suggest a total revision of the English language and style for the entire article using Grammarly or another specialized tool;
-
The paper must follow the specific structure of the journal, namely:
Author Information, Abstract, Keywords, Introduction, Materials and Methods, Results, Discussion, Conclusions, etc., as indicated at: https://www.mdpi.com/journal/sensors/instructions -
You must avoid ending some sections/subsections with equations/formulas or figures (e.g. Table 1 just before subsection 2.2, Figure 3 just before subsection 3.2, Figure 8 just before subsection 4.3, Figure 12 just before subsection 5.3);
-
You must ensure that all figures have the required resolution (minimum 1000 pixels width/height, or a resolution of 300 dpi or higher according to the Journal’s instructions: https://www.mdpi.com/journal/sensors/instructions ). At least Figures 1, 2, 4, 7, and 9 clearly need more resolution;
-
All references to equations/formulas must be explicitly and precisely formulated in the main text (e.g., “see eq. 1” to be included - line 575). The same for eq.2 (“see eq. 2” to be included at lines 578 or 579);
-
In addition, equations should be specified in the Materials and Methods section rather than in the Results section. You should take this principle into account;
-
All digital object identifier (DOI) codes (at least for all journal references) must be explicitly specified;
-
There are many references to conference papers (19 - “conference” and “proc.” as search keys), symposium papers (5), workshops (2 - not for the same items where the “conference” search key occurs), and online materials (also 18 - “Available:” as search key). These mean 44 of all 70 references. In other words, a great majority of non-journal papers. And this is not recommended if you want to publish in a highly rated journal such as Sensors. Additional related and more relevant journal references have to be included in this paper the way that these will become dominant in this work making it more scientifically reliable;
-
There are so many figures (12) in the paper. Some of them (not essential for understanding the main content) should be moved to the Appendix section. If not existing, this section must be created;
-
The title for the Abbreviations section is orphan in relation to the effective list of items. You should move this title to the next page;
-
The last two pages are void and improperly numbered. They should be removed;
-
Even with an existing list of abbreviations, the RAN acronym in the abstract should be briefly explained at the first appearance (in the abstract - line 5).
-
You should provide more details when stating “high security” - lines 4, 245, and 358. For instance, the RSA (Rivest-Shamir-Adleman) encryption will hardly withstand quantum technology attacks in the near future - e.g, https://doi.org/10.1080/23335777.2020.1811384,
https://doi.org/10.3390/electronics11060856, https://link.springer.com/chapter/10.1007/978-981-16-3412-3_2, https://www.technologyreview.com/2019/05/30/65724/how-a-quantum-computer-could-break-2048-bit-rsa-encryption-in-8-hours );
-
In most existing types of virtual machines, the bus/core multiplication factor of the CPU is locked. Or this creates a clear performance disadvantage that you should mention if this applies also to the particular implementations mentioned in this paper (subsection 5.2);
-
Table 3 should provide at least one additional line for comparison purposes (other fields);
-
More details about the ethics agreement must be explicitly specified in such studies based on data collected from human users.
Thank you for your contribution!
Sincerely,
D.H.
Author Response

(The authors gave the same response as above.)

Round 2
Reviewer 3 Report
Dear Authors,
You performed some improvements.
I think the manuscript is closer to the state of being published.
Sincerely,
D.H.